# Immobilization of Radioiodine via an Interzeolite Transformation to Iodosodalite

**DOI:** 10.3390/nano10112157

**Published:** 2020-10-29

**Authors:** Hyejin An, Sungjoon Kweon, Sanggil Park, Jaeyoung Lee, Hyung-Ki Min, Min Bum Park

**Affiliations:** 1Innovation Center for Chemical Engineering, Department of Energy and Chemical Engineering, Incheon National University, Incheon 22012, Korea; hjan_95@inu.ac.kr (H.A.); sjkweon@inu.ac.kr (S.K.); 2Nuclear Energy Team, Lee & Ko, Seoul 04532, Korea; sanggil.park@leeko.com; 3School of Mechanical and Control Engineering, Handong Global University, Pohang 37554, Korea; jylee7@handong.edu; 4LOTTE Chemical Research Institute, Daejeon 34110, Korea

**Keywords:** adsorption, immobilization, interzeolite transformation, methyl iodide, radioiodine

## Abstract

We described a technology for immobilizing radioiodine in the *sod*-cages by the interzeolite transformation of iodine-containing LTA (zeolite A) and FAU (zeolites X and Y) into a sodalite (SOD) structure. The immobilization of iodine in the *sod*-cage was confirmed using diverse characterization methods including powder XRD, elemental analysis, SEM–EDS, ^127^I MAS NMR, and I 3d XPS. Although both zeolites A (Na-A) and X (Na-X) were well converted into SOD structure in the presence of NaI and AgI, the iodide anions were fixed in the *sod*-cages only when NaI was used. The ability to adsorb methyl iodide (CH_3_I) was evaluated for zeolites A and X in which Na^+^ and/or Ag^+^ ions were exchanged, and Ag^+^ and zeolite X showed better adsorption properties than Na^+^ and zeolite A, respectively. However, when both CH_3_I adsorption ability and the successive immobilization of iodine by interzeolite transformation were considered, Na-X was determined to be the best candidate of adsorbent among the studied zeolites. More than 98% of the iodine was successfully immobilized in the *sod*-cage in the SOD structure by the interconversion of Na-X following CH_3_I adsorption, although the Na-X zeolite exhibited half the CH_3_I adsorption capacity of Ag-X.

## 1. Introduction

In the wake of the Fukushima nuclear power plant disaster in 2011, various radioactive elements were released into the environment, raising the public awareness of their danger. Among them, radioiodine is a well known representative radioactive element. It causes human thyroid cancer, since the thyroid easily absorbs iodine. The representative radioiodine isotopes are ^131^I and ^129^I. The former has a short half-life of 8 days and high specific radioactivity. However, ^129^I has a relatively long half-life of 1.6 × 10^7^ years and high mobility in most geological environments, which makes it one of the highest dose contributors [1,2]. When a severe nuclear accident occurs, radioiodine can be released into the atmosphere in the form of organic iodides like methyl iodide (CH_3_I), which accounts for the largest proportion of such radioactive iodide compounds. To selectively remove the highly volatile CH_3_I contained in the gases released from a nuclear accident, dry filtration systems using adsorbents like zeolites can be effectively used to capture CH_3_I and immobile it within the adsorbents [3].

A lot of studies have investigated the adsorptive removal of CH_3_I using various metal ion-exchanged zeolites [4,5,6,7,8,9,10,11]. In evaluations of various metal ion-exchanged Y zeolites (framework type FAU) with similar metal contents, Ag-Y and Cu-Y showed the highest adsorption performance [11]. Compared to other metal iodide compounds, AgI and CuI have much lower solubility products (K_sp_ = 8.5 × 10^−17^ and 1.1 × 10^−12^ at 25 °C, respectively, Appendix A), so they have relatively stronger metal–iodine chemical bonds. Comparisons of CH_3_I adsorption performance over various zeolite structures exchanged with Ag^+^ have also been reported [5,6,7,8]. When CH_3_I is adsorbed on Ag-zeolites, Ag^+^ can combine with I^−^ and be stored in the form of AgI; the CH_3_^+^ binds to the zeolite framework negative charge or extra framework OH^−^ [4,5,9]. Among the various Ag-zeolite structures, Ag-X (FAU) and Ag-Y have exhibited the highest adsorption capacity, and storage in the form of AgI was very effective [6]. On the other hand, the zeolite adsorbents exchanged with other cations like Na^+^ also showed good CH_3_I adsorption ability, although not as good as that of the Ag-zeolites [8,11].

Unlike the other technologies used for the adsorption of harmful substances for removal, it is not necessary to remove the radioactive isotopes by the catalytic conversion and regeneration of the adsorbent. For safety, it is much more important to permanently immobilize the adsorbed radioactive elements. There is some possibility that the metal iodide compounds, formed inside the zeolite structure by adsorbing CH_3_I, can be detached again because the size of metal iodides, e.g., NaI (ca. 3.1 Å) and AgI (ca. 2.8 Å) [7,12], are generally smaller than the size of zeolite pores (<10 Å). It has been reported that the radioiodine can be fixed in a zeolitic structure such as sodalite (SOD) and cancrinite (CAN) by structurally transforming the iodine-containing zeolites by applying high temperature (>800 °C) and high pressure (>100 MPa). This is known as the hot isostatic pressing technique [1,13,14,15,16,17].

Alternatively, some zeolites can be converted into other structures under the mild hydrothermal conditions (<200 °C and <2 MPa), which is called the interzeolite transformation technique [18,19,20,21,22,23,24,25]. It is well known that zeolite structures with lower framework density (FD) have a high potential of converting into a structure with a higher FD, as described by Ostwald’s rule [25]. For example, the zeolite A (LTA) with Si/Al = 1 is easily interconverted to an SOD structure [19], which has structural properties similar to LTA (FD = 12.9 T/1000 Å^3^) but higher FD = 17.2 T/1000 Å^3^ [26]. In addition, it has been reported that the zeolite X with Si/Al = 1.2 (FD = 12.7 T/1000 Å^3^) can also be converted to an SOD structure under certain hydrothermal conditions [23], and converted to other diverse framework structures such as ANA, *BEA, CHA, MAZ, and MER. [18,20,21,22,23,24]. As shown in Figure 1, all the SOD, LTA, and FAU structures have a common sub-structural unit called a *sod*-cage ([4^6^6^8^]). SOD is connected by sharing 4-rings of the *sod*-cage, and the *sod*-cages in LTA and FAU are connected with double 4-rings (*d4r*) and double 6-rings (*d6r*), respectively [26]. Accordingly, LTA and FAU have 8-ring (ca. 4.1 Å) and 12-ring (ca. 7.4 Å) open pores, respectively, allowing small molecules such as CH_3_I and metal iodides to pass through, whereas SOD does not have such open pores. Sodalite cannot efficiently adsorb CH_3_I because it has no open pores, but if radioiodine would be trapped in the *sod*-cage, sodalite can be used as an efficient immobilization substrate.

The SOD structure with Si/Al = 1 has twelve tetrahedral atoms (6Si + 6Al) and two *sod*-cages per unit cell [26]. Thus, in SOD, one *sod*-cage has four cations (e.g., Na^+^) that form a tetrahedral structure around one central anion (e.g., OH^−^ or halogen atoms), and compensate the one central anion, as well as the three negative charges due to the imbalance of the three framework Al atoms [12,14,17,27]. Depending on the type of central anion present in the *sod*-cage, SOD can become hydroxysodalite (HSOD) or iodosodalite (ISOD), which are occupied by OH^−^ and I^−^, respectively [12,14,17]. It has been reported that when the zeolite A (LTA) was used as Si and Al sources together with sodium iodide (NaI), ISOD was formed better than when the general Si and Al precursors like colloidal silica and sodium aluminate, respectively, were used [23]. In addition, ISOD was synthesized using metakaolin, NaI, and NaOH mixed with two types of sodium borosilicate to form pellets, and then heated to a temperature below 850 °C to immobilize iodine [17].

In this study, we first performed the synthesis and characterization of ISOD by the interzeolite transformation of Na-A, Na-X, and Na-Y zeolites in the presence of NaI or AgI. Then, based on these results, the adsorption capacity of CH_3_I was compared to zeolites A and X in which Na^+^ and/or Ag^+^ were ion-exchanged, and successively the immobilization ability of iodine in the *sod*-cage was evaluated by the interzeolite transformation of the CH_3_I adsorbed A and X zeolites to ISOD. All iodide compounds used in this study were nonradioactive.

## 2. Materials and Methods

### 2.1. Synthesis

Na-A (Si/Al = 1.0), Na-X (Si/Al = 1.2), and Na-Y (Si/Al = 2.6) used for the interzeolite transformation and CH_3_I adsorption were purchased from Zeobuilder (Na-A) and Zeolyst (Na-X and Na-Y). The hydrothermal interconversion synthesis of the Na-zeolites was performed with reference to the literature [12], and the details of the synthetic composition are summarized in Table 1. Na-zeolite, NaI (99.5%, Sigma-Aldrich, Saint Louis, MO, USA), and/or AgI (99%, Sigma-Aldrich) were mixed with deionized (DI) water and stirred for 1 h (solution 1). NaOH (50% in H_2_O, Sigma-Aldrich) was diluted in DI water and stirred for 1 h (solution 2). After mixing solutions 1 and 2, the mixture was stirred again for 1 h. The final synthesis mixture was charged into Teflon-lined 23 mL autoclaves and heated at 180 °C for 2 days. If required, pseudoboehmite (99.9%, Sigma-Aldrich) as another Al source was added to the synthesis mixture to modify the Si/Al ratio. The solid products were recovered by filtration, washed repeatedly with distilled water, and dried overnight at room temperature (RT).

The Ag-zeolites were prepared by the ion-exchange of the Na-zeolites, refluxing twice in 0.1 M AgNO_3_ (99.9%, Kojima chemicals Co. Ltd., Saitama, Japan) aqueous solutions (1.0 g solid per 10 mL solution) at 80 °C for 10 h. After the last exchange cycle, the sample was washed with excess DI water (*>*1 L) to eliminate the non-exchanged silver species in the zeolite pores. Before the adsorption experiment, the silver loaded sample was calcined in air at 550 °C for 2 h to uniform the ion sites and desorb the nitrate.

The interzeolite transformation of the CH_3_I adsorbed Na- and Ag-zeolites was performed in the same manner described above. To distinguish between the CH_3_I adsorbed sample and the sample after its interzeolite transformation, the suffixes -ad and -it were added at the end of the sample name, respectively (e.g., Na-X-ad and Na-X-it).

### 2.2. Characterization

Powder X-ray diffraction (XRD) patterns were measured on a SmartLab X-ray diffractometer (Rigaku, Japan) with Cu-Kα radiation in the 2θ range of 3–50° (scan rate = 4° min^−1^). Elemental analysis was carried out on an OPTIMA 7300DV inductively coupled plasma (ICP) spectrometer (PerkinElmer, Shelton, CT, USA). Crystal morphology and average size were determined using a JSM-7800F scanning electron microscope (SEM) (JEOL Ltd., Tokyo, Japan) operating with an acceleration voltage of 15 kV. Electron image acquisitions and elemental analysis were also performed with an energy-dispersive X-ray spectrometer (EDS) coupled with SEM. The 127I MAS NMR spectra were measured on a 500 MHz Avance III HD solid state NMR spectrometer (Bruker, Billerica, MA, USA) at a spinning rate of 8 kHz, a 127I frequency of 100.1 MHz with a π/2 rad pulse length of 2 μs, a recycle delay of 1 s, and an acquisition of ca. 2000 pulse transients. The 127I chemical shift is reported relative to NaI. X-ray photoelectron spectroscopy (XPS) was performed on a PHI 5000 Versa Probe II (ULVAC-PHI 5000 VersaProbe, Japan) with Al-Kα radiation (hν =1486.6 eV).

### 2.3. Adsorption of CH_3_I

Adsorption experiments were carried out in a thermogravimetric analyzer (TGA, Hitachi STA7000) where ca. 10 mg of adsorbent was loaded on a sample holder. Appendix A shows the series of treatments performed on the zeolite adsorbents for the adsorption of CH_3_I following the procedures reported in our previous work [10]. Prior to the experiments, the adsorbent was routinely activated under flowing Ar (99.999%, 100 mL min^−1^) at atmospheric pressure and 200 °C for 1 h and then kept at 100 °C for about 20 min to establish a standard operating procedure, allowing time for the TGA weight balance to be stabilized. A typical design temperature for the dry filtration system is known as RT-200 °C [3]. A mixture of Ar (75 mL min^−1^) and CH_3_I (5000 ppm Ar balance, 50 mL min^−1^), in which the initial concentration of CH_3_I was kept constant at 2000 ppm, was then fed into the TGA for 2 h, and the weight change of the adsorbent was continuously recorded in real time. Here, we used a relatively very high concentration of CH_3_I compared to the practical conditions of a nuclear accident (ca. 0.1 ppm) [3] in order to saturate the zeolite adsorbents with CH_3_I in a reasonable experimental time. After the adsorption for 2 h, the physisorbed CH_3_I was removed at the same temperature under Ar flow for 1 h. Separately, the CH_3_I-adsorbed samples used for interzeolite transformation were obtained in a homemade continuous-flow fixed-bed reactor loaded with ca. 0.5 g of zeolite adsorbents using the same procedure described above.

## 3. Results and Discussion

### 3.1. Interzeolite Transformation to Sodalite

The results of the interzeolite transformation of Na-zeolites into the different structures are summarized in Table 1 (run 1–18). Figure 2a shows the powder XRD patterns for the representative interconverted products obtained after heating at 180 °C for 1 day (run 2, 7, 9, 12, 13, and 17), and the XRD patterns of all the others are depicted in Appendix A. For comparison, the XRD patterns of parent Na-A and Na-X are also displayed in Figure 2a. Since the LTA zeolite with Si/Al = 1 is known to be easily transformed into the SOD structure [19], the interconversion of Na-A was performed under various hydrothermal synthesis conditions (run 1–10). The X-ray peaks for the LTA phase were still observed in the synthesis with ca. 0.7 M of NaOH (NaOH/Al = 1) aqueous solution at 180 °C for 1 day (Appendix A). However, all the LTA phases disappeared and the generation of SOD was confirmed (run 1) after extended crystallization, up to two days. When the NaOH concentration had more than doubled (NaOH/Al = 2–4), a pure SOD phase was crystallized within 1 day (run 2–4). Because they were crystallized under OH^-^ media, these sodalities were HSOD, and their XRD patterns were in good agreement with the literature results [12,19,27]. The I/Al ratio of ISOD with Si/Al = 1 was ca. 0.3, because two iodide anions fit per SOD unit cell [26]. Thus, the interzeolite transformation was performed under the conditions of NaI/Al = 0.3 in the synthesis mixture (run 5–7). Moreover, pure SOD phases were obtained only when the NaOH/Al ratio exceeded two in the synthesis mixture.

Since Ag-zeolites are known to have better CH_3_I adsorption capacity than Na-forms, we tried to perform the interzeolite transformation of Na-A in the presence of AgI instead of NaI (run 8–10). For all the runs, although no zeolitic phases other than SOD were confirmed, the X-ray peaks for bulk AgI clusters and bulk Ag metal phases at 2θ = 22.4°, 23.6°, 39.2°, and 46.2° and 2θ = 38.1° and 44.3°, respectively, were also observed together with the SOD phase. This indicates that unlike NaI, most of the AgI clusters were not dissolved under the hydrothermal condition at 180 °C and existed as a physical mixture together with the generated SOD. However, since the presence of metallic Ag was identified, it can be speculated that some portions of Ag-I were dissociated under the hydrothermal conditions, and the dissociated I^−^ contributed to stabilizing the formation of *sod*-cage of SOD together with Na^+^. On the other hand, the amount of CH_3_I adsorbed using the Ag-zeolite is around 100 mg/g-zeolite [10], which corresponds to 0.1 AgI/Al. So, only 0.1 times as much AgI as Al was added to the synthesis mixture, but the X-ray peaks for AgI and Ag were still observed (run 10). This suggests that the AgI may be difficult to immobilize on the SOD structure by interzeolite transformation, and Na-zeolites may be a better adsorbent from the immobilization point of view, even though their adsorption of CH_3_I is somewhat lower than the Ag-zeolites. Since sodium is cheaper than silver, the former is also more advantageous from an economic point of view.

As described above, the large-pore zeolites like FAU have better CH_3_I adsorption capacity than the small-pore materials like LTA zeolite [6,8]. Accordingly, the interzeolite transformation was performed for Na-X with Si/Al = 1.2, which is similar to that (1.0) of Na-A (run 11–15). The ANA phase was observed as an impurity level together with SOD regardless of the presence or absence of NaI (run 11 and 12). This suggests that the relatively excess amount of silica contributed to the stabilization of ANA phase compared to the interconversion of Na-A [20]. Since FAU and ANA have 6-rings in common, it can be easily converted to ANA by the interconversion of FAU, especially under certain cationic systems like Na^+^ and Cs^+^ [20,21,22]. The ANA structure is not suitable for immobilizing radioactive elements because it has 8-ring open pores (ca. 4.2 Å) [26]. However, when comparing the two strong X-ray peaks of the ANA and SOD phases in the run 12 sample, i.e., 2θ = 26.0 and 24.4 corresponding to (400) and (211) Miller indices for ANA and SOD, respectively, the ANA portion was estimated to be only about 9% of the sample. In contrast, when AgI was added instead of NaI under the identical hydrothermal conditions (run 13–15), although the ANA phase was not observed, the X-ray peaks for bulk AgI and Ag metal phases were also observed, like the results of run 8–10.

Among the zeolite adsorbents, zeolite Y, which has a higher Si/Al ratio (2.6) than zeolite X, is known to exhibit excellent CH_3_I adsorption performance [6,11]. So, additional interzeolite transformation was attempted for Na-Y under the same conditions described above. Only a pure ANA phase was formed, instead of SOD, and bulk AgI and Ag metal phases were also observed in the presence of NaI (run 16) and AgI (run 17), respectively. This can be explained by the much higher silica content, which preferably stabilized the ANA phase rather than SOD. When the interzeolite transformation of Na-Y was attempted with a lower Si/Al ratio of 1.0, by adding more Al precursor, the SOD phase was generated, although not purely (run 18). From the above interzeolite transformation results, it can be concluded that zeolites A and X can be used as CH_3_I adsorbents to form an SOD structure by interzeolite transformation, but zeolite Y is not suitable from the immobilization point of view, even though it has superior CH_3_I adsorption capacity.

Figure 3 (left) shows the SEM images of the representative interconverted products discussed above. The run 7 sample obtained from the interconversion of Na-A in the presence of NaI showed only agglomerated SOD particles ca. 5 μm in size, and there was no impurity phase like NaI. The EDS result also showed that the iodine content on the particle surface was very low, compared to the bulk analysis by ICP (I/Al = 0.01 vs. 0.16 in Table 2). This implies that most of the iodide anions are immobilized in the *sod*-cage in SOD. The SEM–EDS result of the run 12 sample, which was obtained by the interconversion of Na-X with NaI, also showed almost no iodine on the particle surface (Table 2). In addition, as observed in the XRD, a polyhedron impurity crystal about 20 μm in size of something other than SOD particles was observed. This is consistent with the crystal morphology of a typical ANA zeolite in the literature [21]. Meanwhile, the SEM-EDS analysis of the run 9 and 13 samples synthesized in the presence of AgI clearly showed the presence of AgI and/or Ag metal together with SOD particles, which corresponds to the XRD results (Figure 2a).

Figure 4 shows the ^127^I MAS NMR spectra of the representative synthetic products produced by the interzeolite transformation. It is interesting to note that for the run 7 product, no peak was observed at 0 ppm corresponding to NaI, and only a strong ^127^I NMR band was observed at −260 ppm, which is a typical chemical shift for iodine immobilized in an *sod*-cage [12,27]. This strongly indicates that almost all the iodide anions were immobilized in the *sod*-cage of the SOD structure in the run 7 sample, where a pure SOD phase was formed by the interconversion of Na-A in the presence of NaI. For this product, the number of I^−^/*sod*-cage can be estimated to be about 0.5 based on the results of the elemental analysis (Table 2). This means that approximately one iodide anion is immobilized in just one of the two *sod*-cage per SOD unit cell. Furthermore, a strong ^127^I chemical shift was also observed around −260 ppm for the run 12 product, which was synthesized by the interconversion of Na-X in the presence of NaI. For this product, it can be estimated that almost one iodine is immobilized per *sod*-cage in the SOD structure (Table 2). Thus, although the ANA phase co-existed as an impurity level in the run 12 sample, most of the iodine used in the synthesis was immobilized in the *sod*-cage of the mainly generated SOD structure. In contrast, for the run 9 product generated by the interconversion of Na-A with AgI, no peak was observed at −260 ppm, and only a strong band was observed near −70 ppm, which corresponds to bulk AgI clusters (Appendix A). This is also in good agreement with the results from the powder XRD and SEM-EDS analyses. These results indicate that no iodine exists in the *sod*-cages of the SOD structure generated in the run 9 product.

Figure 5 shows the I 3d XPS spectra of representative products synthesized by interzeolite transformation. There are two types of peaks in the ranges of 617–620 eV and 629–632 eV which correspond to I 3d_5/2_ and I 3d_3/2_, respectively [12]. Here, the discussion is based on I 3d_5/2_, because I 3d_3/2_ shows the almost same trend for all samples. For the NaI cluster with the FCC structure, six Na^+^ ions are coordinated around one I^−^ anion, and the XPS peak for I 3d_5/2_ is observed at 618.4 eV [12]. For the run 7 product, the peak for I 3d_5/2_ was observed at 618.7 eV, which is the typical binding energy for I^−^ coordinated with four Na^+^ in the *sod*-cage [12]. We should note here that although the immobilization of iodine in the *sod*-cage of SOD was confirmed in the run 12 product, the XPS peak of I 3d_5/2_ for this sample was observed at 619.5 eV, which is ca. 0.8 eV higher than that of the run 7 product. The binding energy of I^−^ anions can change depending on the type and number of surrounding cations [12,28]. As shown in Table 2, since the run 12 product has the lowest Na/I ratio of 1.8 among the samples analyzed, the strongest I 3d binding energies may be observed. On the other hand, for the run 9 and 13 products, the XPS peak for I 3d_5/2_ was observed at 619.3 eV, which is the typical binding energy corresponding to a bulk AgI cluster [28].

### 3.2. Adsorption of CH_3_I and Immobilization of Iodine in Sod-Cages

Based on the characterization of products obtained from the interzeolite transformation of Na-A, Na-X, and Na-Y in the presence of NaI or AgI, the four different zeolites (i.e., Na-A, Ag-A, Na-X, and Ag-X) were prepared to compare their CH_3_I adsorption properties and the immobilization of iodine by successive interzeolite transformation. As shown in Appendix A, the XRD patterns of the Ag^+^-exchanged pristine Ag-A and Ag-X zeolites displayed no other diffractions like Ag metal or Ag_2_O besides the LTA and FAU phases, respectively [29]. This supports the conclusion that the Ag^+^ ions were mostly dispersed in the zeolite pores and did not diffuse out onto the external surface of the zeolite during the preparation, even after the final high temperature calcination. As shown in Table 3, the Ag contents of the two Ag-zeolites prepared using the identical ion-exchange method are quite similar, and ca. 48% and 43% of the anion sites generated by the framework Al were exchanged with Ag^+^, respectively. From the elemental analysis of commercial Na-A and Na-X, it was confirmed that Na-A (Na/Al = 0.89) contained twice as much Na as Na-X (Na/Al = 0.43).

Figure 6 and Table 3 show the results of CH_3_I adsorption at 100 °C for the four zeolites. While Na-A hardly adsorbed CH_3_I, the other three zeolites were almost saturated after 2 h of adsorption, and the total amount of adsorbed CH_3_I was 140–280 mg/g-zeolite, which is similar to the values reported in the literature [6,10]. Unlike Na-A, the Ag-A showed a CH_3_I adsorption capacity of about 260 mg/g-zeolite. Although the adsorption by Na-X was less than Ag-A, it showed a moderate CH_3_I adsorption capacity of 140 mg/g-zeolite. Ag-X showed about double the CH_3_I adsorption capacity of Na-X. As can be easily predicted from these adsorption results, it thus appears that Ag^+^ is better than Na^+^, and zeolite X is better than zeolite A in terms of CH_3_I adsorption capacity.

As shown in Figure 2b, after CH_3_I adsorption the Na-A-ad and Na-X-ad samples showed no X-ray peaks other than the LTA and FAU phases, respectively. Although no X-ray peaks for bulk AgI clusters were observed in the Ag-A-ad, the characteristic peaks of AgI were observed in the Ag-X-ad. As described above, one of the CH_3_I adsorption mechanisms on the metal sites is the formation of NaI or AgI species, where the successive dissociation of CH_3_-I and the combination of metal iodides occur [4,5,9]. However, XRD detectable AgI may not be formed inside such small zeolite micropores, and therefore AgI clusters or larger particles could form on the external surface of the Ag-X-ad crystallite (Appendix A). It has been reported that a rapid growth of AgI species occurred inside the super cage of the Ag-Y zeolite, and at the same time they diffused out toward the external surface as the spent adsorbent was exposed to ambient conditions [5]. However, since the pore size of LTA is much smaller than that of FAU, it is relatively difficult for AgI formed inside the pores to get out of the pores even after exposure to air, and thus the characteristic AgI X-ray peaks may not be detectable for Ag-A-ad.

The ability to immobilize iodine in the *sod*-cage of the SOD structure by interzeolite transformation was evaluated for the four CH_3_I adsorbed samples discussed above. These synthesis results are also summarized in Table 1 (run 19–22). Although a pure SOD phase (Na-A-it in Figure 2b and Figure 3 (right)) was interconverted from Na-A-ad after only 1 day of crystallization with excessive NaOH concentration (NaOH/Al = 4), no characteristic peaks were observed in the ^127^I MAS NMR (Figure 4) and I 3d XPS (Figure 5) spectra because the amount of CH_3_I adsorption was insignificant, as shown in Figure 6 and Table 3. The Ag-A-it sample also formed an SOD phase under the same interconversion conditions as Na-A-it, but the Ag metal phase was also observed. Interestingly, in this case, unlike the results of interconversion of Na-A with AgI (run 8–10 in Table 1), the bulk AgI phase was not observed. As mentioned above, it can be speculated that AgI molecules or nanometric species formed inside the LTA pores after CH_3_I adsorption may dissociate under the interzeolite hydrothermal conditions, unlike the bulk AgI clusters, and accordingly, the dissociated Ag^+^ may organize a metal cluster. Moreover, the dissociated I^−^ binds with Na^+^ in the synthesis mixture, which can contribute to stabilizing the *sod*-cage of the generated SOD. However, as shown in Figure 4 and Figure 5, in the case of Ag-A-it, the proportion of iodide immobilized in the *sod*-cage was not so high.

Since the interzeolite transformation of FAU may generate the ANA phase with excessive concentrations of sodium as well as silica (run 11, 12, 16, and 17 in Table 1), in the interconversion of Na-X-ad, the concentration of NaOH was lowered to 1/2 compared to those with Na-A-ad and Ag-A-ad (run 21 in Table 1). However, the ANA phase was still formed at a proportion of about ca. 15% compared to SOD (Na-X-it in Figure 2b), which is similar to the result for run 12. We should note here that both the ^127^I MAS NMR (Figure 4) and I 3d XPS (Figure 5) results for Na-X-it were also similar to those of run 12. It thus appears that the iodide anions were immobilized in the *sod*-cage of SOD. The elemental analysis result for Na-X-it was also similar to the run 12 product, and the number of I^−^/*sod*-cage in the SOD structure was estimated to be 0.8, which is slightly less than that (1.0) of run 12 (Table 2).

On the other hand, ANA and Ag metal phases were identified along with the SOD phase from the interconversion of Ag-X-ad with the identical hydrothermal conditions for the Na-X-ad (run 22 in Table 1). Like Ag-A-it, in this run, the bulk AgI phase was not identified (Ag-X-it in Figure 2 and Figure 3 (right)). However, as shown in Figure 4, a weak band for AgI was confirmed around −70 ppm with a strong band at −260 ppm in the ^127^I MAS NMR spectrum of Ag-X-it. This indicates that some of the AgI clusters in Ag-X-ad, identified in the XRD result (Figure 2b), remained even after the interzeolite transformation. Even though Ag-X-ad adsorbed the highest amount of iodine among the four adsorbents (Figure 6 and Table 3), the iodine in Ag-X-it was only half that of Na-X-it (I/Al = 0.13 vs. 0.26 in Table 2), so the number of I^−^/*sod*-cage in SOD was also calculated to be half (0.4 vs. 0.8 in Table 2).

The comprehensive results for CH_3_I adsorption and hydrothermal interzeolite transformation revealed that the Na-form zeolite has a lower CH_3_I adsorption capacity than Ag-form, but a higher iodine immobilization capacity. In addition, unlike Na-A, the interconversion of Na-X to pure SOD is relatively difficult, but the CH_3_I adsorption capacity of Na-X is much higher. Therefore, among the four adsorbents, the Na-X zeolite was determined to be the best adsorbent. In Na-X, the amount of adsorbed iodine measured after CH_3_I adsorption and the continuous desorption of physisorbed CH_3_I was about 120 mg/g-zeolite (i.e., ca. 120,000 ppm) (Table 3). Meanwhile, the concentration of iodine in the Na-X-it solid sample, and the synthetic solution remaining after hydrothermal interzeolite transformation, were determined by analyses to be 117,000 and 2220 ppm, respectively. These results confirmed that the mass balance was well matched within the error range (i.e., 120,000~117,000 + 2220 ppm). Consequently, with the Na-X adsorbent, iodine was immobilized in the *sod*-cage of the SOD structure at a very high proportion of ca. 98%, through a successive interzeolite immobilization process after CH_3_I adsorption.

## 4. Conclusions

A study was carried out to immobilize iodine in *sod*-cages by interzeolite transformation after the adsorption of CH_3_I using Na- and Ag-zeolite adsorbents. Initially, interzeolite transformations were performed for Na-A, Na-X, and Na-Y in the presence of NaI or AgI. It was confirmed that Na-Y, which is known to have the highest adsorption capacity, was not well converted to the SOD structure, unlike the other two zeolites. The CH_3_I adsorption capacity of the four zeolites which can be converted into the SOD structure was then compared under the same conditions. The adsorption ability was found to be in the order of Ag-X > Ag-A > Na-X > Na-A. Therefore, as can be easily expected, Ag^+^ rather than Na^+^ and FAU than LTA had higher adsorption capacity. On the other hand, when the ability to immobilize iodine by the interzeolite transformation of the CH_3_I adsorbed samples was evaluated, although Ag-A showed good adsorption ability and interconverted well into the SOD structure without any impurity, the adsorbed iodine was poorly immobilized in the *sod*-cage. Na-X was less capable of adsorbing CH_3_I than Ag-X but was able to immobilize 98% of the adsorbed iodine. From the overall results, considering both CH_3_I adsorption and immobilization by interzeolite transformation, we demonstrated that Na-X was the most efficient candidate for the removal of radioiodine among the four zeolites evaluated.

## Figures and Tables

**Figure 1 nanomaterials-10-02157-f001:**
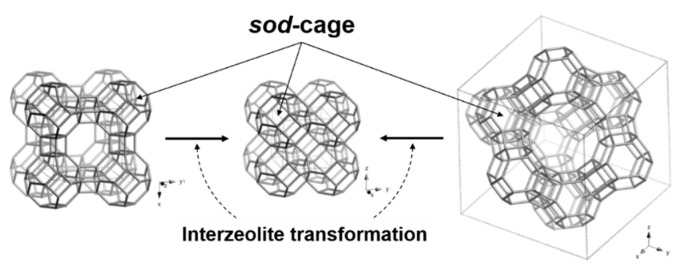
Schematic illustration of the interzeolite transformation of LTA (left) and FAU (right) to sodalite (SOD) (middle).

**Figure 2 nanomaterials-10-02157-f002:**
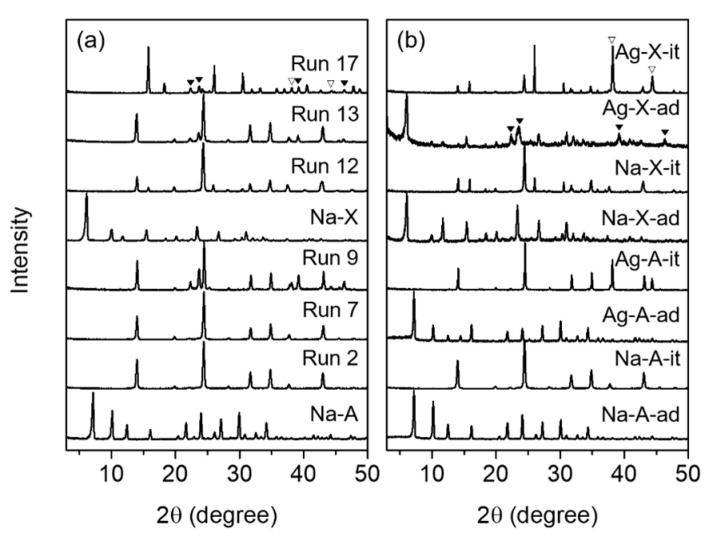
Powder XRD patterns of (**a**) representative products synthesized by the interzeolite transformation of Na-A and Na-X and (**b**) Na-A, Ag-A, Na-X, and Ag-X after the adsorption of CH_3_I and their interzeolite transformation products. For comparison, the XRD patterns of parent Na-A and Na-X are also displayed. The X-ray peaks from AgI and Ag metal are marked by closed (▼) and open inverted triangle (∇), respectively.

**Figure 3 nanomaterials-10-02157-f003:**
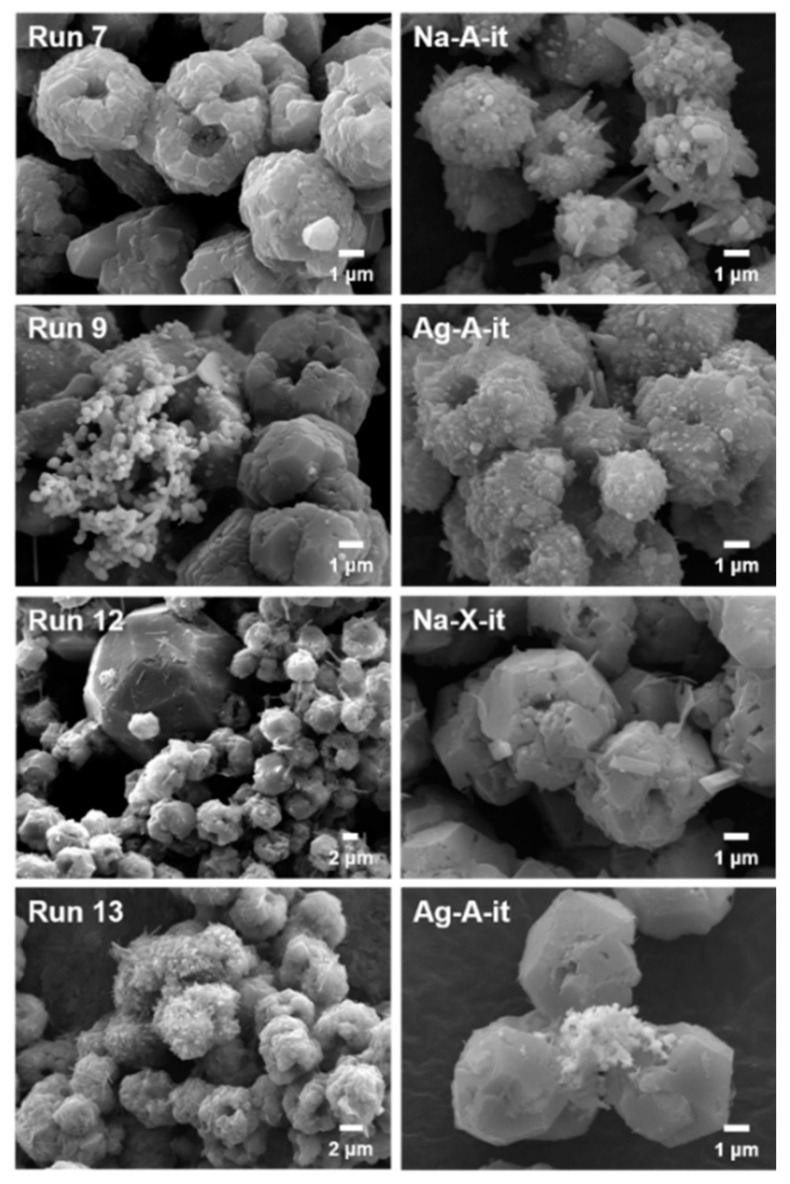
SEM images of the representative products synthesized by the interzeolite transformation of Na-A and Na-X (left), and the interconversion products of CH_3_I adsorbed Na-A, Ag-A, Na-X, and Ag-X (right).

**Figure 4 nanomaterials-10-02157-f004:**
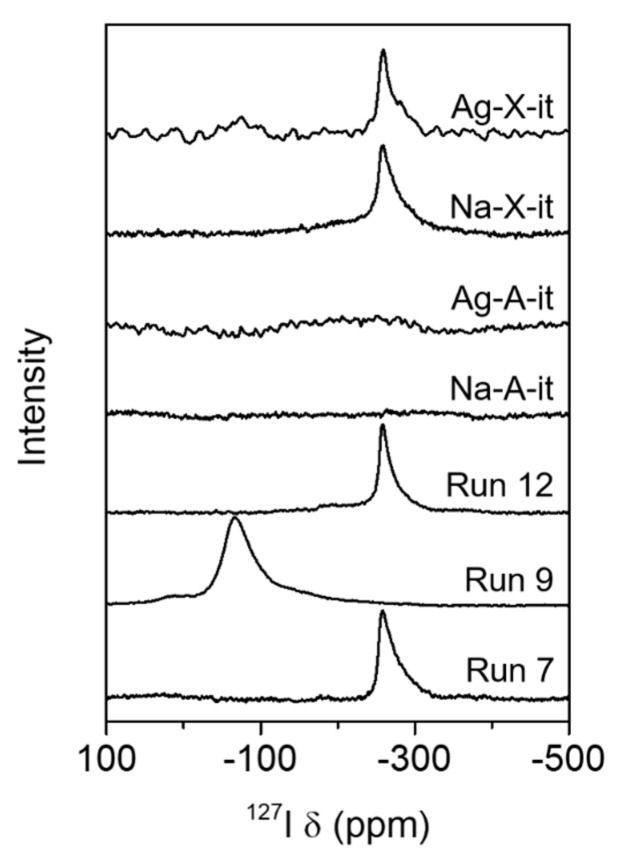
^127^I MAS NMR spectra of the representative products synthesized by the interzeolite transformation of Na-A and Na-X, and the interzeolite transformation products of CH_3_I adsorbed Na-A, Ag-A, Na-X, and Ag-X.

**Figure 5 nanomaterials-10-02157-f005:**
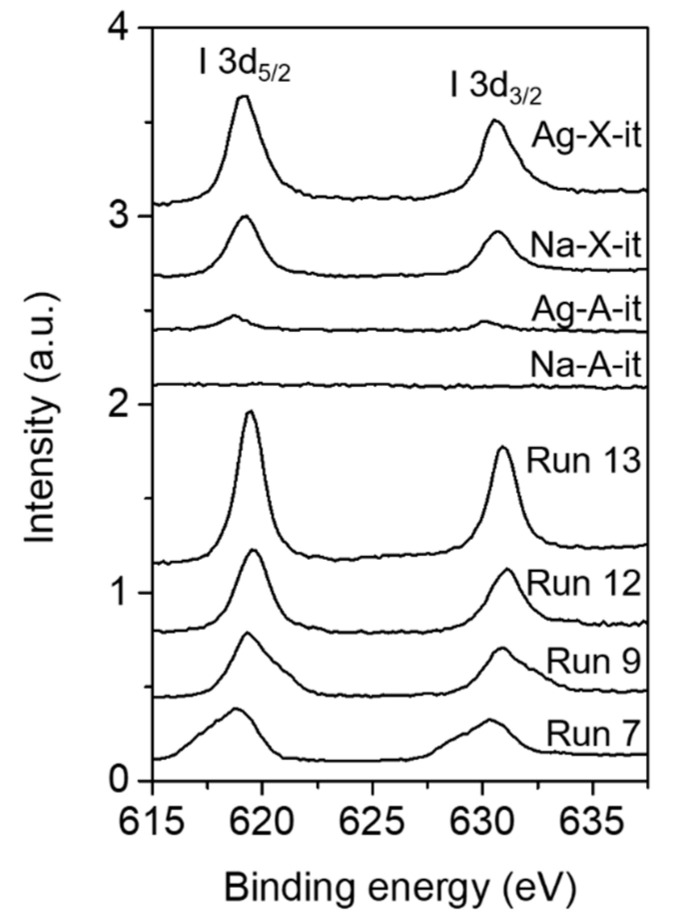
I 3d XPS spectra of the representative products synthesized by the interzeolite transformation of Na-A and Na-X, and the interzeolite transformation products of CH_3_I adsorbed Na-A, Ag-A, Na-X, and Ag-X.

**Figure 6 nanomaterials-10-02157-f006:**
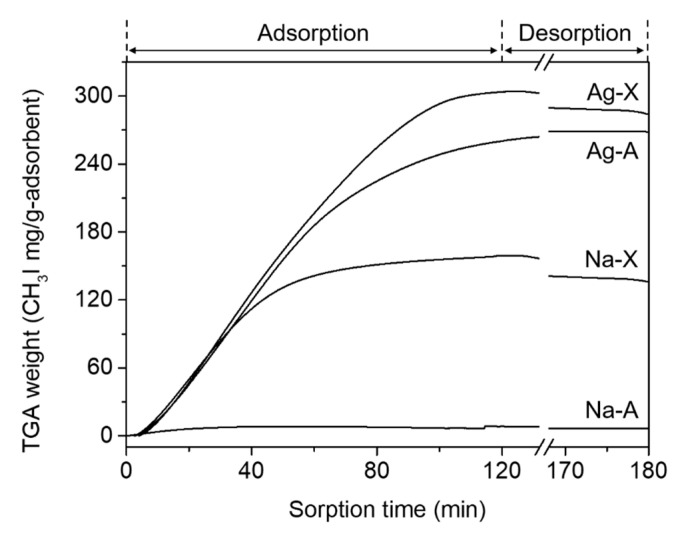
Weight change determined by TGA as a function of time during the adsorption and desorption of CH_3_I on Na-A, Ag-A, Na-X, and Ag-X at 100 °C.

**Table 1 nanomaterials-10-02157-t001:** Representative interconversion hydrothermal synthesis conditions and results.

Run No.	Parent Zeolite ^2^	IZA Code	Synthesis Composition					Product Phase ^3^
			Si/Al	NaOH/Al	MI/Al		H_2_O/Al	
					M = Na	M = Ag		
1	Na-A	LTA	1.0	1	-	-	80	SOD + GIS ^4^
2	Na-A	LTA	1.0	2	-	-	80	SOD
3	Na-A	LTA	1.0	3	-	-	80	SOD
4	Na-A	LTA	1.0	4	-	-	80	SOD
5	Na-A	LTA	1.0	1	0.3	-	80	SOD + GIS ^4^
6	Na-A	LTA	1.0	2	0.3	-	80	SOD
7	Na-A	LTA	1.0	4	0.3	-	80	SOD
8	Na-A	LTA	1.0	2	-	0.3	80	SOD + AgI + Ag
9	Na-A	LTA	1.0	4	-	0.3	80	SOD + AgI + Ag
10	Na-A	LTA	1.0	2	-	0.1	80	SOD + AgI + Ag
11	Na-X	FAU	1.2	2	-	-	80	SOD + ANA + U ^5^
12	Na-X	FAU	1.2	2	0.3	-	80	SOD + ANA
13	Na-X	FAU	1.2	2	-	0.3	80	SOD + AgI + Ag
14	Na-X	FAU	1.2	4	-	0.3	80	SOD + AgI + Ag
15	Na-X	FAU	1.2	4	-	0.1	80	SOD + AgI + Ag
16	Na-Y	FAU	2.6	4	0.3	-	80	ANA
17	Na-Y	FAU	2.6	4	-	0.3	80	ANA + AgI + Ag
18 ^1^	Na-Y	FAU	1.0	2	0.3	-	80	SOD + LTA + U ^5^
19	Na-A-ad	LTA	1.0	4	-	-	80	SOD
20	Ag-A-ad	LTA	1.0	4	-	-	80	SOD + Ag
21	Na-X-ad	FAU	1.2	2	-	-	80	SOD + ANA
22	Ag-X-ad	FAU	1.2	2	-	-	80	SOD + ANA + Ag

^1^ More Al source was added to the synthesis mixture to reach the Si/Al = 1. ^2^ Run 1–18 and 19–22 are the bare and CH_3_I-adsorbed zeolites, respectively. ^3^ All products were obtained after heating at 180 °C for 1 day under static conditions, unless otherwise stated. The product appearing first is the major phase. ^4^ Obtained after heating for 2 days. ^5^ Unidentified phase.

**Table 2 nanomaterials-10-02157-t002:** Chemical composition of the selected products synthesized by interzeolite transformation ^1^.

Product	Na/Al	I/Al ^2^	Na/I	I^−^/*Sod*-Cage ^3^
Run 7	1.1	0.16 (0.01)	7.2	0.5
Run 12	0.60	0.34 (0.00)	1.8	1.0
Na-X-it	0.57	0.26 (0.06)	2.2	0.8
Ag-X-it	0.63	0.13 (0.04)	4.7	0.4

^1^ Determined by inductively coupled plasma (ICP) analysis, unless otherwise stated. ^2^ The values given in parentheses are the I/Al ratios from the SEM–EDS data. ^3^ The number of iodides per *sod*-cage assuming SOD with Si/Al = 1.

**Table 3 nanomaterials-10-02157-t003:** Chemical composition ^1^ and adsorption amounts ^2^ for the four adsorbents studied here.

Adsorbent	Si/Al ^3^	Na/Al	Ag/Al	CH_3_I/Adsorbent (mg/g)	I/Adsorbent (mg/g)
Na-A	1.0	0.89	-	6	5
Ag-A	-	-	0.48	260	230
Na-X	1.2	0.43	-	140	120
Ag-X	-	-	0.43	280	250

^1^ Determined by ICP analysis, unless otherwise stated. ^2^ Determined by TGA after CH_3_I adsorption for 2 h and the consecutive desorption of physisorbed CH_3_I for 1 h. ^3^ Provided by the vendor.

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
