# Peer review of "Immobilization of Radioiodine via an Interzeolite Transformation to Iodosodalite"

_nanomaterials, 2020, doi:10.3390/nano10112157_

Round 1
Reviewer 1 Report
I want to convey to the team that participated in this manuscript that the work is very good and very topical.
Author Response
We would like to thank this reviewer not only for carefully reading our manuscript, but also for giving us very positive comments.
Reviewer 2 Report
The objective of the manuscript is relevant in the radionuclide waste management and the experiments have been well planned. However, the text need a major revision.
In general, the organization of the results and the manuscript structure make difficult to follow the scientific content, which is quite interesting. Moreover, the description of the results are insufficient and the discussion and conclusion are not supported enough by the results. It should be convenient that the results were analysing in deep and the manuscript was re-written.
Other particular items are:
1.- The title is confusing because they do not perform the immobilization of radioiodine, they explore the possibility to use a mechanism of immobilization but with their stable isoptopes
2.- Line 92-96: The authors described the adsorption of CH3I on zeolite A, X and Y, and afterword transforms those zeolites in sodalite. Why do not they try to use directly sodalite?
3.- Experimental (line 99): Authors do not justify de synthesis of those zeolites with those Si/Al ratio
4.- Adsorption of CH3I: Figure S1, Why did the author chose those treatment conditions?
5.- Interzeolite Transformation to Sodalite: The XRD results has not been well explained they have been only discussed. A description of the patterns is recommended.
6.- line 216-220: a comparison between I/Al ration at surface and in the bulk is performed. It is quite clear that the particle surface composition has been extracted from EDS results but it is not clear the technique used for the bulk composition.
Reviewer 3 Report
The manuscript is well-written and well-organized that addresses an important topic related to the immobilization ability of iodine in the sod-cage obtained by the interzeolite transformation of Na-A, Na-X, and Na-Y zeolites in the presence of NaI or AgI. The paper gives a good survey of the issue, potential nuclear power plant disaster, and the various radioactive elements released into the environment.
Author Response

(The authors gave the same response as above.)

Round 2
Reviewer 2 Report
The athors have modified the text but they have not appropiately answered the questions
Author Response
We are sorry that our responses were not sufficient to satisfy this reviewer, but we did our best to revise the manuscript. We would like to thank again this reviewer for careful reviewing our manuscript.